# Validity and Reproducibility of Food Group-Based Food Frequency Questionnaires in Assessing Sugar-Sweetened Beverage Consumption Habits among Chinese Middle-School Students

**DOI:** 10.3390/nu15183928

**Published:** 2023-09-10

**Authors:** Junyao Yi, Guoye Song, Zhenghao Lin, Yuting Peng, Jieshu Wu

**Affiliations:** 1School of Public Health, Nanjing Medical University, Nanjing 211166, China; jyyi@stu.njmu.edu.cn (J.Y.); sgy@stu.njmu.edu.cn (G.S.); 2Department of Maternal, Child and Adolescent Health, School of Public Health, Nanjing Medical University, Nanjing 211166, China; 13023462171@163.com (Z.L.); pennynjmu@njmu.edu.cn (Y.P.)

**Keywords:** food frequency questionnaire, sugar-sweetened beverages, food group, validation, reproducibility, adolescents, sugar intake, obesity, Chinese

## Abstract

Assessing the intake of sugar-sweetened beverages (SSBs) is crucial for reducing obesity; however, a simple but relatively accurate method for determining added sugar consumption among school adolescents is lacking. The aim of this study was to evaluate the reproducibility and validity of a food group-based food frequency questionnaire (FG-FFQ) for SSBs in assessing SSB consumption and added sugar among middle-school students. A total of 242 school students completed the FG-FFQs twice and four discontinuous 24-h dietary records (24HDR) over a three-month period. A weighted average approach was used to obtain the average sugar content in the sugary drink food group (FG). Correlation coefficient, weighted kappa statistic, misclassification analysis, and Bland–Altman plot were used to evaluate the validity and reproducibility of the FG-FFQ. Linear regression was utilized to obtain the calibration formulas. The average content of added sugar in sugary drink FG was 8.1 g/100 mL. SSB consumption frequency, consumption amount, and added sugar had correlation coefficients of 0.81, 0.87, and 0.87, respectively, in the validity analysis (*p* < 0.05). The majority of scatter plots were covered by 95% confidence intervals in the Bland–Altman bias analysis. The intra-class correlation coefficient of SSB consumption frequency and Spearman correlation coefficient of SSB consumption amount and added sugar were 0.74, 0.81, and 0.90, respectively, in the reproducibility analysis (*p* < 0.05). Results produced by the FG-FFQ calibration formula were more comparable to 24HDR. The FG-FFQ for SSB consumption showed acceptable validity and reproducibility, making it a viable instrument for epidemiological studies on sugary drinks in adolescents.

## 1. Introduction

The global obesity epidemic is not optimistic [1]. According to population-based research worldwide, the prevalence of obesity among children and adolescents aged 5–19 years increased from <1% in 1975 to 6–8% in 2016 [2]. By 2025, it is estimated that 206 million children and teenagers aged 5–19 years will be obese [3], with China having the greatest number [4]. Notably, mounting evidence suggests that eating unhealthy foods, such as sugar-sweetened beverages (SSBs), contributes to adolescent obesity. Each serving of SSB consumed daily increases children’s BMI by 0.05 kg/m^2^ annually [5], which may be a time bomb for future increases in BMI-related global mortality and other disease burdens [4]. Consequently, the World Health Organization (WHO) recommends reducing SSB intake in youngsters to prevent childhood obesity [6]. To achieve this goal, the accurate identification and screening of individuals with high SSB consumption in an extensive population is essential.

Epidemiological studies frequently employ a 24-h dietary recall/record (24HDR) and food frequency questionnaire (FFQ) to assess dietary intake [7]. However, these approaches have different advantages and practical issues [8,9,10]. The 24HDR can provide reliable dietary information but does not reflect long-term eating habits. The FFQ is used to understand consumption habits, but its results are affected by food categorization. Moreover, regardless of the method used, sufficient time for recall and the cooperation of respondents are critical for reproducibility. SSBs are available in a wide variety, and categorizing them appropriately and opportunely is challenging. Conversely, the over-classification of SSBs increases students’ memory burden and decreases their cooperation rates. The time-consuming survey method is also especially challenging for obtaining student, parental, and school support. Therefore, a simple yet relatively accurate way to obtain information on SSBs and added sugar consumption among teenagers is urgently needed.

Owing to the resource-intensive nature of dietary survey methods and the indispensability of dietary information for medical research, food group-based food frequency questionnaires (FG-FFQs) have become popular. Notably, FG-FFQs focus on a group of foods that share certain characteristics (such as high sugar, high fat, and high salt) or are associated with certain health outcomes [11,12,13], to explore the relationship between foods and hypertension, cancer, chronic kidney disease, and others [12]. Added sugar is the culprit for the health risks of SSBs [14], the contents of which in various SSB types fluctuate within a relatively stable range owing to its flavor, and it can be considered a food group (FG). This may contribute to streamlining the laborious processes of categorization and statistics, lessening the memory bias of teenagers, and lowering the likelihood of incorrect classification. Therefore, we hypothesized that the FG-FFQ for SSB is effective at surveying students’ SSB consumption and assessed the validity and reproducibility of an FG-FFQ method for sugary drinks to investigate SSB consumption and added sugar intake among middle-school students, using four discontinuous investigator-assisted 24HDR for 3 days as the standard [15].

## 2. Materials and Methods

### 2.1. Study Design and Population

This research was conducted at a middle school in Suzhou City, Jiangsu Province, China, that was willing to cooperate. Two classes in each grade from 7 to 9 in the school were selected to participate in this study. Students’ demographic information (grade and gender) and the first FG-FFQ (FG-FFQ 1) survey were investigated in class under the guidance of the researcher and teacher in charge. Height was measured in cm without shoes using a wall-mounted rangefinder, and weight was calculated using a digital scale. For validity research, 114 volunteer students were recruited for a 3-day 24HDR four times. Three months later, the FG-FFQ (FG-FFQ 2) was administered once more to all students in the six classes. Students who completed both FG-FFQ surveys were included in this reproducibility study. Students who participated in the validity study were excluded from the reproducibility data analysis. The students were not informed in advance that they would complete the questionnaire twice. The flowchart of the study is depicted in Figure 1.

### 2.2. SSB Assessment through FG-FFQ

Currently, there is no unified definition of SSBs. SSBs in this study were defined as all non-diet and non-alcoholic drinks with added sugars that provide calories, whether packaged or manufactured, referring to the 2015–2020 Dietary Guidelines for Americans [16]. The SSBs types listed in the FG-FFQ included tea drinks, sweetened fruit juice, coffee drinks, energy/sports drinks, milk beverages, vegetable juice, carbonated drinks, sweet milk teas, low-calorie and vegetable protein beverages, which correspond to those listed in the 15-item beverage intake questionnaire (BEVQ-15) [17]. Sweet milk teas and probiotic drinks (as milk beverages) were included because of their popularity in China. These two types of drinks are mistakenly considered healthy because of their milk and probiotic ingredients, despite the high sugar content of up to 12.0 and 12.5 g/100 mL in sweet milk tea and Yakult beverages, respectively. Milk, soymilk, 100% fruit juice, and drinks with sugar substitutes were excluded as they did not contain additional sugars.

SSB consumption through FG-FFQ was assessed by the question “How often have you consumed SSBs, excluding 100% fruit juice and sugar substitute drinks, in the past 1 month?”, followed by lists of detailed categories of beverages included and excluded and the brands or names of common beverages. After fully considering the logical tangent point, the answer options were set to “daily”, “1–3 times a week”, “4–6 times a week”, “1–3 times a month”, and “less than 1 time a month”, so that students can recall the consumption times in the past period of time and select the corresponding range, avoiding recall or filling difficulties. For consumption amount assessment, the questionnaire included questions pertaining to portion size “Referring to the quantity indicated in the picture, how much each time would you normally drink SSBs over the past month?” A series of auxiliary images were used to describe and quantify the reference volumes of popular packaging in the market, such as cans and bottles, to support and enhance the memory of the respondents. Total consumption was calculated by multiplying the median value of the frequency by the daily portion size of the SSB.

### 2.3. SSB Assessment through 24HDR

The 24HDR is often used as the gold standard for dietary assessment [18]. To avoid the seasonal variation, we used a 24HDR at baseline and performed it once a month for the following 3 months. The students were required to keep track of three day separate 24-h diets that were randomly assigned on two weekdays and one weekend each month, taking into account dietary variations throughout the week. A professional researcher validated the beverage consumption of the students telephonically by asking them to write down all the food and drink they consumed in the previous 24 h. Weekly sugary drink consumption was calculated using a weighted ratio of 5:2 for weekdays and weekends (total beverage consumption was quantified as the sum of all beverages), and the average daily intake of SSBs was calculated.

Each SSB consumed was photographed, and its name, type, and amount were recorded. The content of added sugar in the SSB was determined using the nutritional facts label in the pictures. In addition to the ten types of sugary beverages listed in the definition of SSBs, substitute sugar and natural sugar drinks were also involved in analysis, whereas milk, fermented milk, alcoholic beverages, and tea were not. Based on the above 12 types of beverages, three weighted average sugar contents of the sugary drink FG (all added sugar intake through SSBs divided by the total volume of the SSBs) were calculated to assess the added sugar consumed through the beverages. The three average sugar contents are from different types of beverages below: (1) Common added sugar beverages. (2) All added sugar beverages. (3) All sweet-tasting beverages (including sugar-substituted beverages and 100% fruit juices, considering student recall bias).

### 2.4. Statistical Analyses

Statistical analyses were performed using statistical analysis software (SPSS v. 26.0 Windows, 10, SPSS Inc., Chicago, IL, USA). The normality of data in the confirmability and reproducibility studies was tested using the Shapiro–Wilk test. Owing to the biased distribution of SSB consumption frequency and amount, the data are described as medians and quartiles.

We calculated the r-values of the Spearman correlation coefficient (SCC) for SSB consumption frequency and the intra-class correlation coefficient (ICC) for SSB consumption and added sugar. The following definitions are used to explain r-values and ICC: (1) r (ICC) ≤ 0.40 was considered “poor”, (2) 0.41 < r (ICC) < 0.59 was considered “fair”, (3) 0.60 < r (ICC) < 0.74 was considered “good”, and (4) 0.75 < r (ICC) < 1.00 was considered “excellent” [19]. Kappa statistics were used to evaluate gross classification. SSB consumption and sugar intake were divided into four categories according to quartiles, which are commonly used in the estimation of the proportion of misclassifications in dietary intake validation studies. Bland–Altman plots were used to assess the consistency of SSB consumption amount and added sugar. The above indicators were also used to evaluate reproducibility, except for the median differences.

To obtain more accurate intake data through FG-FFQ, linear regression models were used to derive the calibration factors α and β for three indicators. SSB consumption frequency, consumption amount, and added sugar were calculated by considering FFQ- and 24HDR-estimated SSB consumption as the independent and dependent variables, respectively. The formula used is as follows: Calibrated dietary intakes = α + β FG-FFQ, where, α: regression constant; β: slope of regression; and FG-FFQ: Dietary intakes as estimated through FG-FFQ [20]. For all indicators, *p* values < 0.05 were considered statistically significant.

## 3. Results

A total of 283 students participated in the survey, among which, 38 were eliminated because they completed the FG-FFQ only once (not in school on the survey day). We recruited 114 out of 283 students to participate in our validity study, and three students were excluded due to incomplete 24HDR information, with a response rate of 97.4%. Finally, the reproducibility and validity analyses included data from 131 and 111 students, respectively. In 242 students ultimately included in analysis, males (48.8%) and females (51.2%) were evenly distributed. Students in grades seven, eight, and nine comprised 35.5, 23.6, and 40.9% of the total, respectively. BMI of the participants was 20.8 (95% CI: 19.1, 22.3), with 12.4 and 9.9% overweight and obese students, respectively.

We collected 659 beverage intake records from the 24HDR data, including 415, 11, and 9 records from SSB, substitute sugar and natural sugar drinks, respectively. According to the contribution analysis, among the 415 consumption records of SSBs shown in Table 1, 95.9% (398/415) of the consumed types were centered on 6 out of 10 types of beverages, accounting for 97.4% of the total amount of added sugar. Therefore, we defined these six types as common added sugar beverages. The median sugar level of the six types of SSBs ranged from 6.0 to 10.9 g/100 mL, with the variation within the type ranging from 0 to 5.5 g/100 mL (Table 1). The intragroup and intergroup variation ranges of sugar content in various types of SSBs are similar. The three weighted average sugar contents of sugary drink FG obtained based on this were 8.6, 8.1, and 7.8 g/100 mL, respectively, which were used to examine the reproducibility and validity of the FG-FFQ in assessing added sugar intake.

The consumption frequency and amount were calculated. Then the corresponding added sugar results were obtained according to above three average sugar content of sugary drink FG. Table 2 shows a comparison of the consumption frequency, daily SSB consumption amount, and added sugar obtained using the FG-FFQ and 24HDR and the correlation between them during the FG-FFQ validation phase. The findings revealed that the relative differences in the three indicators were 13.3, 6.1, and 0.5% (for 8.1 g/100 mL), with modest median differences, and the correlation coefficients were 0.81, 0.87, and 0.87 (for 8.1 g/100 mL), indicating a strong and significant association between FG-FFQ and 24HDR (*p* < 0.001). See Table 2 for details. Furthermore, >90% of the participating students were classified into the same or adjacent (±1) quartile. The weighted kappa values were 0.32, 0.42, and 0.40 (for 8.1 g/100 mL), demonstrating moderate agreement across all groups (*p* < 0.001). The results are summarized in Table 3.

A Bland-Altman bias analysis was performed to examine the agreement between the two methods, with deviations plotted against the means. As indicated in Figure 2a,b, the 95% confidence interval covered the majority of the scatter plots of SSB and sugar intake from SSB, demonstrating that the level of agreement between the two methods was considered acceptable. In contrast, the FG-FFQ overestimated SSB consumption and added sugar by an average of 146.1 mL/week and 13.5 g/week, respectively. Furthermore, as the intake increased, a slight proportional bias for greater differences between the FFQ and 24HDR was observed. The limit of agreement was large because of the substantial standard deviations in the differences.

Table 4 presents the reproducibility of the two FG-FFQs. For SSB consumption frequency, total SSB consumption, and added sugar, the ICC and SCC correlation coefficients were all >0.7 (*p* < 0.001), indicating a highly positive correlation. The weighted Kappa values were 0.41 (*p* < 0.001) for consumption frequency and 0.56 (*p* < 0.001) for consumption amount and added sugar, demonstrating moderate agreement between the two measurements. Most participating students (>85%) were in the same or adjacent quartiles. Similar reproducibility results were obtained for the different added sugar contents calculated using the three methods. Bland–Altman analysis showed that <10% of participating students were outside the acceptable limit of the protocol and that there was a positive mean difference in SSB consumption and added sugar, as shown in Figure 2c,d.

The calibration coefficients (α and β), calibrated mean consumption frequency, consumption amount of SSB, and added sugar are presented in Table 5. The β values of the three variables were essentially the same, with added sugar and consumption frequency showing the lowest (0.62) and highest (0.68) β values, respectively. These coefficients revealed that for consumption frequency, SSB consumption amount, and added sugar, the mean calibrated values of the FG-FFQ were comparable to those of the test method (24HDRs) (Table 5).

## 4. Discussion

Imminent control of SSB consumption requires a quick and reasonably accurate tool for screening high-risk populations and monitoring intervention effects, especially for students. In this school-based study, the SCC, ICC, and Bland–Altman plots showed good validity and reproducibility, while kappa values and misclassification showed acceptable validity and reproducibility consistency, suggesting that this short FG-based semi-quantitative beverage consumption questionnaire method is acceptable. In addition, the average sugar content of the sugary drink FG and the degree of bias in the FG-FFQ assessment of SSB consumption were explored.

The concept of FG has emerged in recent years and has been applied in whole-grain food and hypertensive diet health assessment research [12,19]. The FFQ has been widely employed in large epidemiological studies to examine the association between the frequency of sugary beverage consumption and chronic diseases, such as obesity and metabolic syndrome. The sweetened beverages in the FFQ are divided into three to seven items for investigation and then merged into one category for analysis [21,22,23]. The FG-FFQ for SSB focuses on a broader range of categories and can include more sugar-contributing beverages, which simplifies the procedure and makes it prone to obtaining intuitive data. However, the reproducibility and validity of the previous FFQ focused on nutrients [24], and the assessment of sugary drinks or added sugar consumption among students is lacking.

The median difference, ICC, SCC, cross-classification, kappa value, and Bland–Altman analysis were comprehensively employed in this study to evaluate the validity of the FG-FFQ for SSB consumption. The results showed that compared with 24HDR, the median difference for FG-FFQ was insignificant for SSB consumption frequency and consumption amount, with correlation coefficients of 0.81 and 0.87, respectively, indicating excellent consistency. The FG-FFQ for SSB has yet to be evaluated. Only a few studies examined the validity of FFQ for SSB, with Huybrechts finding a Spearman correlation coefficient of 0.503 in children aged 2.5 to 6.5 years [25] and Vereecken and Maes finding one of 0.46 in adolescents aged 11 to 12 years [26]. Our findings are better than those of the aforementioned studies, which could be because the children in the previous study were all under the age of 12, whereas our students were older, more cooperative, and more accurate in reporting their beverage intake. This could also be related to the fact that we employed multiple discrete 24HDR to better reflect long-term consumption. However, the validity of an FFQ should not be determined using correlation coefficients alone. Notably, in this study, <10% misclassification was observed according to Lombard’s summary of identified statistical tests and interpretation criteria for the validation of dietary intake assessment methods [27], indicating a high level of agreement. Weighted Kappa values were likewise acceptable and superior to those of Huybrechts’ study [25]. Moreover, the Bland–Altman analysis used to evaluate consistency revealed that the FG-FFQ and 24HDR were significantly consistent and widely comparable. Nonetheless, the FG-FFQ marginally overestimated SSB intake (146.1 g/week), especially when evaluating high consumption. However, it can be adjusted through the obtained calibration coefficients.

In the reproducibility assessment, the consumption frequency and amount had correlation coefficients (ICC and SCC) that were both >0.7, similar to the findings of BEVQ-15 (r = 0.74) by Fausnacht et al. [17]. A misclassification of 2.3% is better than the 17% serious misclassification observed in the study conducted by Vereecken and Maes using a 15-item FFQ for school children (<12 years) [26], possibly due to the older age and reduced recall burden of the students in our study. With no appreciable deviation between the two FFQ calculations, the Bland–Altman plots showed that the FG-FFQ had an excellent ability to evaluate individuals’ SSB consumption over a longer period. The consumption of SSBs was severely misclassified as slightly >10% (10.7%), which may be related to the long interval of 3 months. A single consumption amount may change with seasonal alternation (winter to spring in our survey) even when the consumption habits (consumption frequency) of sugary drinks do not change considerably.

Studies on SSBs have focused on the sugars added to them. However, the measurement of added sugar in SSBs is complicated because of the wide variety of beverages and the variability in sugar content among them [14]. Notably, >95% of the beverages consumed in this study contained commonly added sugars, and there were similar differences in sugar levels between and within the beverage groups. These findings provide a theoretical foundation for application of FG-FFQ to SSBs. After obtaining the average sugar content of the SSBs, we first assessed the reproducibility and validity of the FG-FFQ for measuring added sugar in SSBs. No significant differences in the median intake of added sugars in SSBs were found between the FG-FFQ and 24HDR, the correlation (>0.7) and agreement were good, which are higher than the results of only a handful of studies using the FFQ (49–204 food items) combined with food composition databases to assess the dietary intake of added sugars, such as fructose and sucrose (r > 0.3) [28,29,30]. This may be linked to the rise in mistakes caused by the over-categorization of foods in other research, and it also justifies the evaluation of a single FG. After evaluation, all three average sugar contents had good reproducibility and validity in evaluating sugar intake, but the evaluation results considering all sweet drinks were more in line with the 24HDR findings. The results suggest that even though we explicitly stated in the question that SSBs do not include natural sugars and sugar-substitute beverages, and provided examples, students may still mistakenly count them because they have the same sweet taste. This may require attention in the future when the proportions of natural sugars and sugar-substitute beverage consumption are high.

The WHO has long advocated limits for added sugar consumption in SSBs owing to their health risks [31]; however, the assessment of added sugar intake has been challenging. Traditional survey techniques have not been widely promoted among students and have become a significant contributing factor to the ineffective management of SSBs in adolescents [24]. This study established that the FG-FFQ has good validity and reproducibility for evaluating middle-school students’ SSB consumption habits. The method is quick and simple to use, might lessen the burden on children’s memories, and is preferred by parents, schools, and students. Further, this method has been used many times for evaluating SSB consumption among Chinese middle-school students. The questionnaire can be completed on paper or electronically and is supplemented by corresponding examples to explain the types and graphic images to indicate the volume of sugary drinks. Compared with plain text or tabulated questionnaires, it is more fascinating and scientific and consequently receives more compliance from students. In addition, we determined the calibration coefficients of the FG-FFQ for SSB consumption and added sugar assessments to obtain results that were more closely aligned with the 24HDR. The calibrated median SSB intake of 1.5 times/week indicated in this survey is lower than that of Canadian students of the same age group [32], who consumed 1.5 times/day. However, the prevalence of obesity among Chinese adolescents is comparable to that of Canada [33]. The FG-FFQ for SSB approach can easily and accurately assess students’ SSB consumption, allowing it to screen out high SSB consumers in large populations of pupils and then implement targeted intervention. Meanwhile, it can be used repeatedly throughout the intervention to track the results and then enable prompt adjustment of intervention strategies, which will greatly aid in controlling students’ consumption of SSB and subsequently lessen the burden of obesity and disease with important public health implications.

However, this method has certain limitations. First, recall bias is inevitable in retrospective investigations. Second, the approach solely measured the added sugar in SSB and did not include sugar in other foods. Studies have shown that sugar in SSB is more likely to have negative health effects than solid food consumption, owing to less satiation with SSB [34,35]. Furthermore, teenagers who drink more SSBs tend to consume more other sweet items, indicating that the consumption of SSBs reflects adolescents’ preference for sweet items [36]. Additionally, there are significant ethnic, socioeconomic, and cultural differences in the use of SSBs between and within nations; therefore, paying attention to proper adaptation when using SSBs in various cultural contexts is crucial.

## 5. Conclusions

In conclusion, the FG-FFQ has acceptable reproducibility and validity and can be used as a reliable tool for epidemiological studies on SSB in adolescents. As more populations and SSB types continue to evolve, future evaluations and revisions of the method are needed to improve their adaptability to multiple populations, cultural contexts, and eras.

## Figures and Tables

**Figure 1 nutrients-15-03928-f001:**
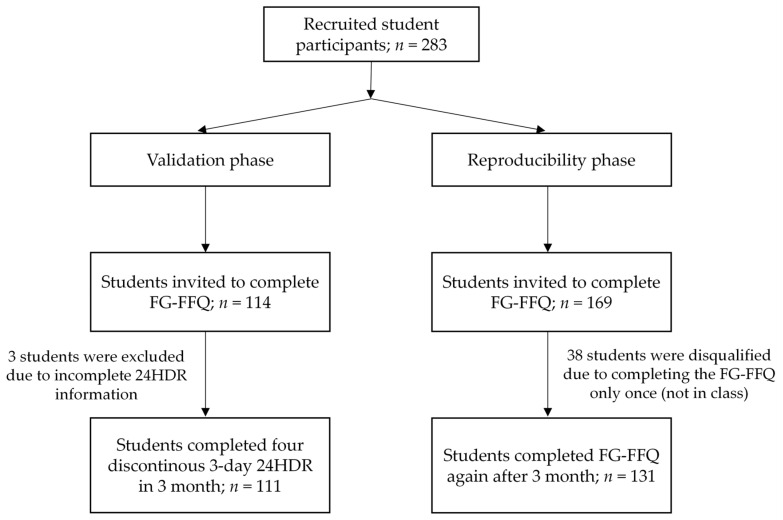
Flowchart of participant selection and reliability and validity evaluation.

**Figure 2 nutrients-15-03928-f002:**
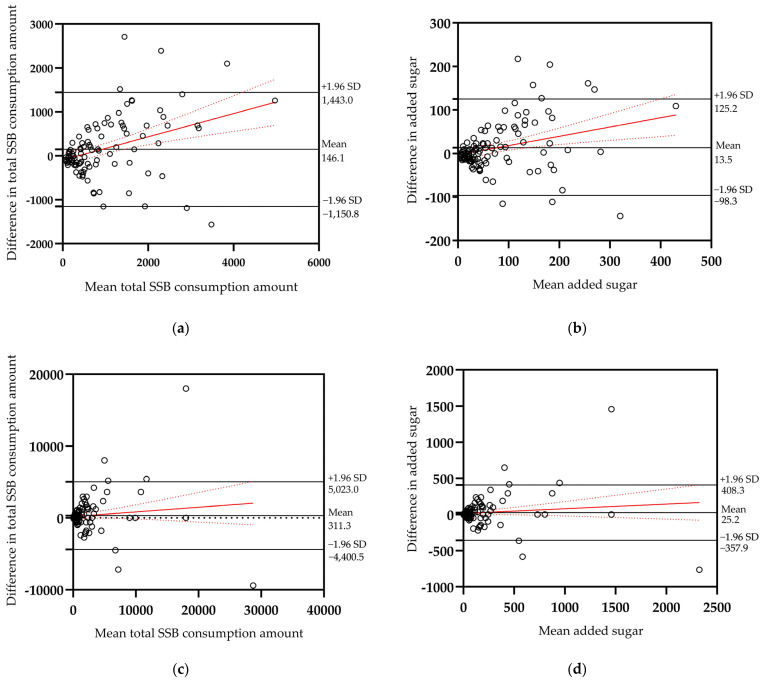
Bland-Altman plots for validity of total SSB consumption amount (g) (**a**), validity of added sugar (g) (**b**), reproducibility of total SSB consumption amount (g) (**c**), reproducibility of added sugar (g) (**d**). The difference between the mean estimate of SSB consumption amount and added sugar by two dietary assessment methods (*y*-axis) was plotted against the mean of SSB consumption amount and added sugar measured by two dietary assessment methods (*x*-axis).

**Table 1 nutrients-15-03928-t001:** Added sugar contents and the contribution to added sugar intake of different types of SSBs.

Types of Beverages	*n* ^1^	Added Sugar Content (g/100 mL)		Single Sugar Intake (g)		Sugar Intake for Each Type of SSB (g)
		Median (P_25_–P_75_)	IQR	Median (P_25_–P_75_)	IQR	Amount (%)
Carbonated drinks	132	10.9 (8.6–10.9)	2.3	27.3 (17.3–35.0)	17.7	3603.6 (34.8)
Sweetened fruit juice	81	9.7 (9.0–10.3)	1.3	25.8 (17.8–32.9)	15.1	2089.8 (20.2)
Sweet milk tea	68	7.1 (7.1–8.0)	0.9	35.5 (28.4–40.0)	11.6	2414.0 (23.3)
milk beverages	53	6.0 (4.5–10.0)	5.5	12.5 (11.3–18.0)	6.7	662.5 (6.4)
Vegetable protein beverages	37	7.0 (7.0–7.0)	0	17.5 (14.0–28.0)	14.0	647.5 (6.2)
Tea drinks	27	8.9 (21.1–38.6)	17.5	25.0 (14.5–21.1)	6.6	675.0 (6.5)
Energy/sports drinks	5	4.8 (4.8–5.0)	0.2	21.6 (16.8–24.0)	7.2	108.0 (1.0)
Vegetable juice	5	5.0 (5.0–5.5)	0.5	14.5 (7.5–15.0)	7.5	72.5 (0.7)
Low-calorie beverages	4	1.8 (1.8–2.0)	0.2	5.4 (5.2–7.4)	2.2	21.6 (0.2)
Coffee drinks	3	4.7 (4.7–4.7)	0	14.1 (12.9–16.5)	3.6	66.3 (0.6)

^1^ Number of consumption records.

**Table 2 nutrients-15-03928-t002:** Comparison of medians and correlations between the FG-FFQ and the 24HDR (*n* = 111).

Variables	FG-FFQ Median (P_25_–P_75_)	24HDR Median (P_25_–P_75_)	MD ^1^	%MD ^2^	Correlation Coefficient ^3^
consumption frequency (times/week)	1.5 (0.5–4.5)	1.7 (0.9–3.2)	−0.2	13.3	0.81 ^4^
total SSB consumption amount (mL)	495.0 (200.0–1485.0)	525.0 (250.0–1109.5)	−30	6.1	0.87 ^5^
added sugar ^6^ (g)	44.3 (17.7–150.5)	44.1 (21.5–87.3)	0.2	0.5	0.87 ^5^
added sugar ^7^ (g)	44.3 (17.7–141.8)	44.1 (21.5–87.3)	0.2	0.5	0.87 ^5^
added sugar ^8^ (g)	44.3 (17.6–136.5)	44.1 (21.5–87.3)	0.2	0.5	0.86 ^5^

^1^ MD: Median difference = median FG-FFQ—median 24HDR. ^2^ Percentage of median difference was computed using the formula: |median FG-FFQ—median 24HDR|/median FG-FFQ × 100%. ^3^ *p* < 0.001. ^4^ Spearman correlation coefficient. ^5^ intra-class correlation coefficient. ^6^ Calculated based on an average sugar content of 7.8 g/100 mL for the sugary drink FG. ^7^ Calculated based on an average sugar content of 8.1 g/100 mL for the sugary drink FG. ^8^ Calculated based on an average sugar content of 8.6 g/100 mL for the sugary drink FG.

**Table 3 nutrients-15-03928-t003:** Comparison of cross-classification between the FG-FFQ and the 24HDR (*n* = 111).

Variables	Cross-Classification into Quartiles; *n* (%)			Kappa ^3^
	Classified into Same Quartile	Classified Adjacently ^1^	Grossly Misclassified ^2^	
consumption frequency (times/week)	51 (46.0)	51 (46.0)	9 (8.1)	0.32 **^4^**
total SSB consumption amount (mL)	63 (56.8)	40 (36.0)	8 (7.2)	0.42 **^5^**
added sugar ^6^ (g)	64 (57.7)	41 (36.9)	6 (5.4)	0.44 **^5^**
added sugar ^7^ (g)	61 (55.0)	44 (39.6)	6 (5.4)	0.40 **^5^**
added sugar ^8^ (g)	61 (55.0)	44 (39.6)	6 (5.4)	0.40 **^5^**

**^1^** Classified into the same or adjacent (±1) quartile. **^2^** Classified into opposing g quartiles (by ≥2 quartiles). **^3^**
*p* < 0.001. **^4^** Spearman correlation coefficient. **^5^** intra-class correlation coefficient. ^6^ Calculated based on an average sugar content of 7.8 g/100 mL for the sugary drink FG. ^7^ Calculated based on an average sugar content of 8.1 g/100 mL for the sugary drink FG. ^8^ Calculated based on an average sugar content of 8.6 g/100 mL for the sugary drink FG.

**Table 4 nutrients-15-03928-t004:** Reproducibility results of two FG-FFQs—comparison of medians, misclassification, coefficients, and Kappa (*n* = 131).

Variables	Median (P_25_–P_75_)		Cross-Classification into Quartiles; *n* (%)			Kappa ^3^	Correlation Coefficient ^3^
	FG-FFQ 1	FG-FFQ 2	Classified into Same Quartile	Classified Adjacently ^1^	Grossly Misclassified ^2^		
consumption frequency (times/week)	2.0 (0.5–6.0)	2.0 (0.5–6.0)	91 (69.5)	37 (28.2)	3 (2.3)	0.56	0.74 ^4^
total SSB consumption amount (mL)	800.0 (250.0–2400.0)	800.0 (150.0–1600.0)	73 (55.7)	44 (33.6)	14 (10.7)	0.41	0.81 ^5^
added sugar ^6^ (g)	62.4 (187.2–19.5)	62.4 (11.7–124.8)	73 (55.7)	44 (33.6)	14 (10.7)	0.41	0.90 ^5^
added sugar ^7^ (g)	64.8 (20.3–194.4)	64.8 (12.2–129.6)	73 (55.7)	44 (33.6)	14 (10.7)	0.41	0.90 ^5^
added sugar ^8^ (g)	68.8 (21.5–206.4)	68.8 (12.9–137.6)	73 (55.7)	44 (33.6)	14 (10.7)	0.41	0.90 ^5^

^1^ Classified into the same or adjacent (±1) quartile. ^2^ Classified into opposing g quartiles (by ≥2 quartiles). ^3^ *p* < 0.001. ^4^ Spearman correlation coefficient. ^5^ intra-class correlation coefficient. ^6^ Calculated based on an average sugar content of 7.8 g/100 mL for the sugary drink FG. ^7^ Calculated based on an average sugar content of 8.1 g/100 mL for the sugary drink FG. ^8^ Calculated based on an average sugar content of 8.6 g/100 mL for the sugary drink FG.

**Table 5 nutrients-15-03928-t005:** Calibration parameters and median (P_25_–P_75_) for SSB consumption estimated from FG-FFQ, 24HDR and calibrated FG-FFQ for students (*n* = 111).

Variables	α (95% CI)	β (95% CI)	FG-FFQ Median (P_25_–P_75_)	24HDR Median (P_25_–P_75_)	Calibrated FG-FFQ Median (P_25_–P_75_)
consumption frequency (times/week)	0.48 (0.11–0.85)	0.68 (0.58–0.78)	1.5 (0.5–4.5)	1.7 (0.9–3.2)	1.5 (1.0–2.9)
total SSB consumption amount (mL)	225.33 (91.45–359.20)	0.63 (0.54–0.72)	495.0 (200.0–1485.0)	525.0 (250.0–1109.5)	535.2 (351.3–1160.9)
sugar intake ^1^ (g)	16.34 (4.33–28.35)	0.66 (0.56–0.76)	44.3 (17.7–150.5)	44.1 (21.5–87.3)	45.6 (28.0–106.4)
sugar intake ^2^ (g)	16.43 (4.30–28.56)	0.65 (0.55–0.75)	44.3 (17.7–141.8)	44.1 (21.5–87.3)	45.2 (28.0–108.6)
sugar intake ^3^ (g)	16.72 (4.39–29.06)	0.62 (0.53–0.72)	44.3 (17.6–136.5)	44.1 (21.5–87.3)	44. 2 (27.7–110.0)

^1^ Calculated based on an average sugar content of 7.8 g/100 mL for the sugary drink FG. ^2^ Calculated based on an average sugar content of 8.1 g/100 mL for the sugary drink FG. ^3^ Calculated based on an average sugar content of 8.6 g/100 mL for the sugary drink FG.

## Data Availability

Data will be made available on request.

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
