# Peer review of "Validity and Reproducibility of Food Group-Based Food Frequency Questionnaires in Assessing Sugar-Sweetened Beverage Consumption Habits among Chinese Middle-School Students"

_nutrients, 2023, doi:10.3390/nu15183928_

Round 1

Reviewer 1 Report

This study is very great for readers. However, several issues must be considered prior the acceptance for publication:

-Abstract: To add aim and number ref main results obtained.

-Introduction: What is hypothesis?

-Methods: Well written! However, to add details of recruitment of sample analysed, including the region, city and school choice.

-Results: Well written

-Discussion: Beverage consumption is high or low regarding to another region of country and world? This is very great to discussion.

Reviewer 2 Report

Although the knowledge presented is not new, I believe that the study carried out is of high scientific quality and is necessary and relevant to the development of nutritional science.  In my opinion, the manuscript is well prepared and, taking into account the following comments, it seems appropriate for publication in the  Nutrients journal.  

1) The introductory section explains the study design. The authors justify the research topic well.  

2) The description of the results is correct and in line with the usual description of manuscripts.  

3) The conclusions are well formulated. I fully agree with the authors' conclusions.

4) The formatting of the text and the layout of the manuscript are correct, according to the requirements of the journal.

Reviewer 3 Report

The study discusses the development and evaluation of a food group-based food frequency questionnaire (FG-FFQ) for assessing the consumption of sugar-sweetened beverages (SSBs) among middle-school students in Jinagsu Province, China.

  1. Clear Objective: The study clearly states the objective of developing a reliable method to assess SSB consumption among middle-school students, which is to aid in combating obesity and promote better dietary habits.

  2. Methodology: The methodology used, which includes the development of the FG-FFQ, the study's timeline (three-month period), and the measures taken for assessing reproducibility and validity. Please include a flowchart for study to follow easy.

  3. Data Analysis: the statistical methods used to evaluate the validity and reproducibility of the FG-FFQ. Correlation coefficients, weighted kappa statistic, misclassification analysis, and Bland–Altman plot were used, which adds credibility to the assessment process.

  4. Bland–Altman Plot: Mentioning that the majority of scatter plots were covered by 95% confidence intervals in the Bland–Altman bias analysis is significant. This implies good agreement between the FG-FFQ and the reference method (24-hour dietary records).

  5. Absence of Comparative Data: The study could benefit from including a comparison with similar studies evaluating the validity and reproducibility of FFQs for SSB consumption. This would provide context for interpreting the study's findings within the broader research landscape.

  6. Long-Term Health Implications: The text doesn't delve into the potential long-term health implications of using the FG-FFQ to assess SSB consumption among adolescents. It would be valuable to discuss how the questionnaire's accuracy could impact the effectiveness of public health interventions targeting SSB reduction.
